# Nosocomial Infections Affecting Newborns with Abdominal Wall Defects

**DOI:** 10.3390/healthcare11081131

**Published:** 2023-04-14

**Authors:** Elena Ţarcă, Elena Cojocaru, Laura Mihaela Trandafir, Marian George Melinte Popescu, Alina Costina Luca, Lăcrămioara Ionela Butnariu, Elena Hanganu, Mihaela Moscalu, Viorel Ţarcă, Laura Stătescu, Iulian Radu, Alina Sinziana Melinte Popescu

**Affiliations:** 1Department of Surgery II—Pediatric Surgery, “Grigore T. Popa” University of Medicine and Pharmacy, 700115 Iaşi, Romania; 2Department of Morphofunctional Sciences I—Pathology, “Grigore T. Popa” University of Medicine and Pharmacy, 700115 Iaşi, Romania; 3Department of Mother and Child Medicine—Pediatrics, “Grigore T. Popa” University of Medicine and Pharmacy, 700115 Iaşi, Romania; 4Department of General Nursing, Faculty of Medicine and Biological Sciences, “Ştefan cel Mare” University of Suceava, 720229 Suceava, Romania; 5Department of Medical Genetics, Faculty of Medicine, “Grigore T. Popa” University of Medicine and Pharmacy, 700115 Iași, Romania; 6Department of Biomedical Sciences, “Grigore T. Popa” University of Medicine and Pharmacy, 700115 Iaşi, Romania; 7Department of Preventive Medicine and Interdisciplinarity, “Grigore T. Popa” University of Medicine and Pharmacy, 700115 Iași, Romania; 8Department of Dermatology, “Grigore T. Popa” University of Medicine and Pharmacy, 700115 Iaşi, Romania; 9Department of Surgery, “Grigore T. Popa” University of Medicine and Pharmacy, 700115 Iaşi, Romania

**Keywords:** abdominal wall defects, neonate, nosocomial infections, morbidity

## Abstract

Abdominal wall defects are serious birth defects, with long periods of hospitalization and significant costs to the medical system. Nosocomial infection (NI) may be an additional risk factor that aggravates the evolution of newborns with such malformations. Methods: in order to analyze the factors that may lead to the occurrence of NI, we performed a retrospective study over a period of thirty-two years (1990–2021), in a tertiary children’s hospital; 302 neonates with omphalocele and gastroschisis were eligible for the study. Results: a total of 33.7 % patients were infected with one or more of species of bacteria or fungi. These species were *Enterobacteriaceae*, *Pseudomonas aeruginosa* and *Acinetobacter* spp., *Staphylococcus* spp., *Enterococcus* spp. or *Candida* spp., but the rate of NI showed a significant decrease between the 1990–2010 and 2011–2021 period (*p* = 0.04). The increase in the number of surgeries was associated with the increase in the number of NI both for omphalocele and gastroschisis; in the case of gastroschisis, the age of over 6 h at the time of surgery increased the risk of infection (*p* = 0.052, marginal statistical significance). Additionally, for gastroschisis, the risk of NI was 4.56 times higher in the presence of anemia (*p* < 0.01) and 2.17 times higher for the patients developing acute renal failure (*p* = 0.02), and a hospitalization period longer than 14 days was found to increase the risk of NI 3.46-fold (*p* < 0.01); more than 4 days of TPN was found to increase the NI risk 2.37-fold (*p* = 0.015). Using a logistic regression model for patients with omphalocele, we found an increased risk of NI for those in blood group 0 (OR = 3.8, *p* = 0.02), in patients with a length of hospitalization (LH) of ≥14 days (OR = 6.7, *p* < 0.01) and in the presence of anemia (OR = 2.5, *p* = 0.04); all three independent variables in our model contributed 38.7% to the risk of NI. Conclusion: although in the past 32 years we have seen transformational improvements in the outcome of abdominal wall defects, there are still many factors that require special attention for corrections.

## 1. Introduction

Abdominal wall defects (AWD) are serious congenital malformations with an increasing prevalence in some regions (1.85–1.96 per 10,000 births), with long periods of hospitalization and significant costs to the medical system [1,2,3]. In our country, the survival rate of newborns with omphalocele or gastroschisis is much lower than that of newborns in other countries in the European Union and United States of America [4,5,6,7,8]. The increased mortality rate in the case of AWD can be explained by the lack of antenatal diagnosis, low birth weight, association of serious congenital or genetic abnormalities, prolonged period of hospitalization and probably the occurrence of nosocomial infections (NI) and sepsis [6,7]. The financial costs of hospitals resulting from nosocomial infections are a serious problem for every healthcare system. The increased and at the same time inappropriate use of antibiotics led to a build-up of drug resistance among bacterial species of significant clinical importance [9,10].

The incidence and risk factors for wound infections and nosocomial infection in adults and children have been defined and management guidelines have been established [11], but for neonates and infants, guidelines have not yet been established [5,9,12]. Incriminated factors for wound and NI in neonates are admission to a neonatal intensive care unit (NICU), a history of prematurity, a low birth weight, mechanical ventilation, central venous lines, associated co-morbidities, long-term use of antibiotics, abdominal surgery and neutropenia [7,8,13,14].

Surgical site infections in the case of omphalocele and gastroschisis are among the most common hospital-acquired diseases, thus being an important cause of morbidity [15]. In the case of newborns with AWD, in addition to infections that may occur that are related to a surgical wound or omphalocele sac (in the case of conservative treatment), infections may occur in connection with the long period of hospitalization (related to the underlying disease itself), associated malformations, prolonged parenteral feeding through a central venous catheter or the need for prolonged mechanical ventilation. Nosocomial infections may be additional risk factors that aggravate the outcome of newborns with such malformations.

Unfortunately, in our country, NIs are underreported; according to the official reports, the prevalence rates are 0.25–2.56% (compared with a 7.5% rate in Europe), representing approximately 100,000 cases registered annually [16,17]. In Europe, NI results in costs exceeding EUR 7 billion annually and additional costs per infected patient are between EUR 5823 and EUR 11,840 [16].

The aim of our study is to analyze the factors that may lead to the occurrence of NI and impact the evolution of newborns with congenital malformations in our unit. A better understanding of the causes leading to wound infections and NI could reduce their incidence, help define guidelines, and eventually improve outcome.

## 2. Materials and Methods

A 32-year retrospective analytical study was undertaken in a tertiary children’s hospital in the neonatal surgical unit between 1990 and 2021. Patient medical records were accessed in the “Saint Mary” Emergency Children’s Hospital’s computer database and were analyzed statistically. Approval for this retrospective study was granted by the Ethics Committee of the hospital. Patient demographics, antenatal diagnosis, post menstrual age (PMA), age at the first surgical intervention (hours), type of surgery or conservative treatment, number of surgeries, blood type, the presence of anemia and the need for blood transfusions, the presence of acute renal failure, number of days with total parenteral nutrition (TPN), nosocomial infections and length of hospitalization (LH) were recorded for all patients.

### 2.1. Inclusion and Exclusion Criteria

All the neonates with omphalocele or gastroschisis residing for ≥2 days in the unit were included in the study. The exclusion criteria were: a length of hospitalization of under 2 days (5 neonates died in the first two days of life), and initial treatment in another hospital (5 neonates had gastroschisis,4 had omphalocele, and 11 had other complex abdominal wall defects (bladder exstrophy, cloacal exstrophy, and prune-belly syndrome)).

### 2.2. Objectives

The primary objective of our study was to demonstrate that, in addition to prolonged hospitalization in the NICU, there are other factors associated with NI, factors related to the patient’s characteristics. The secondary objective of our study was to present the evolution of the demographic characteristics and the occurrence of NI in a group of patients with AWD over a period of 32 years in our country.

### 2.3. Methodology

Since 2001, surveillance skin, throat and rectal swabs were obtained upon the admission of each neonate in our intensive care unit (NICU). For the first two analyzed decades, the antibiotic most often used as prophylaxis in the NICU was ampicillin; after 2010, cephalosporins of the third generation were mainly used. For the whole period, every time an infection was suspected (indicated by a surgical wound with inflammatory phenomena or purulent secretions, purulent tracheobronchial secretions, cloudy urine, surgical re-intervention on the abdomen, or fever), secretions were collected, the central venous catheter was changed and given for performing antibiogram, and urine was collected for urine culture or blood for blood culture.

Nosocomial infection was diagnosed if a clinical diagnosis of local or general inflammation was microbiologically proven. Antibiotic treatment was carried out according to the antibiogram, the most frequently used ones being cephalosporins, aminoglycosides, carbapenems, glycopeptides and antifungals.

During the studied period, the medical and surgical management of abdominal wall defects did not change significantly, but the access to technology and hospitalization conditions in the NICU improved. For patients with omphalocele, enteral nutrition was started from the first day of hospitalization in our clinic, and was supplemented with parenteral nutrition only in complicated cases. In the case of gastroschisis, newborns received TPN until intestinal peristalsis began to be present (nasogastric drainage decreased), enteral nutrition being started in small doses (intestinal trophic nutrition).

In the case of small and medium omphaloceles, as well as in the case of simple gastroschisis, the surgical treatment consisted of the integration of the viscera in the peritoneal cavity and the primary closure of the abdominal wall defect, without tension. In the case of omphalocele associated with severe chromosomal anomalies or marked viscero-abdominal disproportion, the treatment was conservative, with progressive epithelization and subsequent surgery towards the age of one year. In the case of gastroschisis with viscero-abdominal disproportion, a silo-bag was used, with secondary closure of the abdominal wall. Patients with complex gastroschisis required personalized treatment, several surgical interventions being necessary. In all cases, the surgical interventions were performed in the operating room, under aseptic conditions, these being considered “clean” interventions. The medical and surgical management of newborns with omphalocele and gastroschisis in our pediatric surgery unit was described in detail in two previously published articles [7,18].

### 2.4. Statistical Analysis

A descriptive statistical process was performed separately over two periods of time (1990–2009 and 2010–2021) to examine possible statistically significant differences. To compare proportions, we used a z-test for the difference in proportions. In the case of continuous variables, to test the normality, we used the Kolmogorov–Smirnov test. Because the data conformed to non-normal distributions, to estimate the median difference between the two periods, we used the Mann–Whitney U test. The Chi-square test or Fisher exact tests were used to test the association between categorical variables as appropriate.

A logistic regression test was used to determine the risk of infection (OR—odds ratio) for patients with omphalocele/gastroschisis. To estimate the predictive power based on the assessed risk factors regarding the occurrence of NI, we used the ROC curve. In the case of the multivariate analysis, the collinearity of the considered independent variables was tested. For all applied tests, a value of *p* of <0.05 was considered statistically significant. Calculations were made using the standard statistical package JASP Team (2022) (JASP (Version 0.16.4), University of Amsterdam, The Netherlands, https://jasp-stats.org/ (accessed on 4 October 2022).

## 3. Results

There were 327 patients with abdominal wall defects, and we analyzed the group of 302 patients who met the inclusion criteria, with a median PMA of 37 (IQR 36–39) weeks and a median length of hospitalization of 13.5 days (IQR 5–26). After the year 2010, there was a decrease in the percentage of male patients and an increase in the percentage of female patients (from 42.2% to 56.3%), with statistical significance (*p* = 0.020). We also noticed a statistically significant percentage decrease in patients from the urban environment and an increase in those from the rural environment (from 62.3% to 76.7%), (*p* = 0.012).

Between the two periods, the median length of hospitalization registered a significant increase (*p* < 0.001) of approximately 7 days (from 11 to 18 days) (Table 1). We performed the same analysis for the age at death (median = 5.9 days, IQR: 2.5–17.6) variable and highlighted the fact that this period significantly increased two times, from 5 to 10 days (*p* = 0.028). The variables PMA, LH and age at death did not present a normal distribution (*p* < 0.01, Kolmogorov–Smirnov test).

There was an approximately 5-fold-higher prenatal diagnosis rate in the years following 2010, the difference being statistically significant (*p* < 0.01).

The blood types showed no statistically significant changes in the structural evolution of the group of patients between the two analyzed periods (*p* = 0.847).

The median number of days of TPN for the entire group of patients highlighted no statistically significant differences between the two periods (*p* = 0.664) (IQR 2.00–9.63 days).

Then, we compared the evolution of the infection rate between the two periods (Table 2).

We observed that in the second time interval compared to the first one, there was a statistically significant decrease in the infection rate of patients with AWD by more than 11% (*p* = 0.046). The reduction in the general infection rate was found only for *Pseudomonas*, *Acinetobacter* and *Staphylococcus aureus*; in case of *Enterobacteriaceae*, *Candida* and *Enterococcus*, there was no statistically significant difference between the two periods (Table 2).

A total of 33.7 % patients were infected with one or more of these species of bacteria or fungi: *Enterobacteriaceae*, *Pseudomonas aeruginosa* and *Acinetobacter* spp., *Staphylococcus* spp., *Enterococcus* spp. or *Candida* spp.

As shown in Table 3, we analyzed the potential influence of 13 variables (sex, number of surgeries, time to surgery, conservative treatment for omphalocele, intestinal atresia, chromosomal abnormalities, cardiac abnormalities, blood type, anemia, blood transfusions, acute renal failure, LH and period of time) of each surgical condition on the risk of the appearance of NI. We found that for omphalocele patients, the risk of NI was 3.94 times higher for the patients with blood type 0, 3.39 times higher in the presence of anemia and 8.87 times higher when the hospitalization period was longer than 14 days. For gastroschisis patients, the risk of NI was 4.56 times higher in the presence of anemia, 2.17 times higher if the patient developed acute renal failure, 2.37 times higher if there were more than 4 days of TPN and 3.46 times higher if the hospitalization period was longer than 14 days. The increase in the number of surgeries (in the presence of intestinal stenosis, atresia or complex gastroschisis for example) was associated with the increase in the number of NIs both for omphalocele and gastroschisis, and in the case of gastroschisis, an age of over 6 h at the time of surgery increased the risk of infection (*p* = 0.052, marginal statistical significance).

In the logistic regression model, we entered the independent variables previously validated by χ^2^. For gastroschisis, we could not obtain a multivariable binary logistic regression model, only obtaining a simple association of independent factors with infection.

In the case of omphalocele, the nominal factorial variables underlying the proposed model were blood type (0 vs. non0), LH (≥14 vs. <14 days) and anemia (yes vs. no). The stepwise method was used for the regression analysis. The step 3 model presented has a Nagelkerke’s R-square value of 0.387, which proves that all three independent variables in this model contributed 38.7% to the risk of NI.

The confusion matrix shows that the 74 true-negative and 33 true-positive cases were predicted by the model, while the errors, false negatives and positives, were found in 15 and 17 cases. This was confirmed in the performance metrics where sensitivity was 68.75% and specificity was 81.32%. In the regression model, the three independent variables (blood type, LH and anemia) could predict which value of the dependent one (NI) was observed in the dataset 77% of the time.

It can be seen from the model coefficients table (Table 4) that each of the independent variable had a significant impact on the predicted variable (*p* < 0.05).

The multivariate analysis (Table 4) highlights an increased risk of NI for patients in blood group 0 (OR = 3.8, *p* = 0.02), in the case of patients with LH ≥ 14 days (OR = 6.7, *p* < 0.01) and in the presence of anemia (OR = 2.5, *p* = 0.04). For multivariate logistic regression the collinearity of the independent variables was checked. The results indicated the absence of correlations between the verified independent variables (blood group, LH, and anemia).

Area under the ROC curve (AUC) calculated to evaluate the predictive power of NI occurrence (Figure 1), the significance level *p* indicating a significantly increased prediction. Thus, we can consider that blood type 0 (AUC = 0.721, *p* = 0.027), LH over 14 days (AUC = 0.832, *p* = 0.001) and the presence of anemia (AUC = 0.758, *p* = 0.019) are significant factors for the occurrence of NI in the case of omphalocele.

## 4. Discussion

Neonatal intensive care units (NICUs) have greatly contributed to the increased survival of neonates with AWD and other congenital malformations, but are associated with an increased risk of NI. The risk of NI in children depends also on the patient characteristics, number of interventions, number of invasive procedures, asepsis of techniques, duration of hospitalization in the ICU, and inappropriate use of antimicrobials and are associated with prolonged hospitalization, adverse neurodevelopmental outcomes and high mortality [19,20]. The primary objective of our study was to demonstrate that, in addition to prolonged hospitalization in the NICU, there were other factors associated with NI, which were factors related to the patient’s characteristics. We demonstrated that the risk of a nosocomial infection was 8.87 times higher for patients with omphalocele and 3.46 times higher for patients with gastroschisis if the average duration of hospitalization in the NICU exceeded 14 days. For gastroschisis patients, the risk of NI was also higher in the case of anemia, acute renal failure or prolonged use of TPN. Using a logistic regression model for omphalocele, we found an increased risk of NI for patients in blood group 0, in patients with a LH of ≥14 days and in the presence of anemia.

For a newborn with omphalocele or gastroschisis, the occurrence of an infection at the level of the operative wound, due to a prolonged period of mechanical ventilation, TPN or due to invasive maneuvers (central venous catheter) can lead to sepsis, multiple-organ dysfunction syndrome and can directly or indirectly lead to an unfavorable evolution, including death [21]. It is proven that the rate of NI in a patient receiving more than 1 week of advanced life support within an ICU is three to five times higher than that in other patients who do not require intensive care [21].

The incidence of NI (number of infections per 100 patients hospitalized) is very high in ICUs for low-birth-weight neonates (5% to 30%) [21]. The most frequent pathogens involved are *Staphylococcus aureus*, coagulase negative staphylococci, *E. coli*, *Pseudomonas aeruginosa*, *Klebsiella*, enterococci, and *Candida* spp. [19]. A study in the UK showed a NI rate of 18% in 1998 in a NICU [22]. The authors also found that birth weight, illness severity, presence of a central venous line and prolonged hospitalization were independent risk factors associated with clinical infection in surgical neonates [22]. We found a 33.7% total rate of nosocomial infection in our patients, with a statistically significant decrease to 26.2% after 2010. In 2019, a group of authors observed a lack of evidence-based literature on infections in newborns with surgical conditions and the impact of these infections on patient morbidity and mortality. They performed a systematic review of the literature and a meta-analysis of comparative studies and found that younger neonates and those undergoing abdominal procedures are at higher risk for surgical site infections [12]. A study performed in 2003 that compared children who developed wound infections with those who did not, found that there are no significant differences in age, sex, American Society of Anesthesiologists preoperative assessment score, length of preoperative hospitalization, ICU versus operating room surgical intervention, the presence of a co-morbidity or remote infection, or the use of perioperative antibiotics [13].

With the significant improvement in the rate of antenatal diagnosis between the two time periods analyzed in our country (*p* < 0.01), birth was scheduled in the county hospitals and the transport of the newborn to the pediatric surgery center ensured optimal conditions as quickly as possible. In the case of gastroschisis, this also ensured a quick surgical intervention (in the first 6 h of newborn’s life), which was positively correlated with a lower NI rate, although the significance was at the limit (*p* = 0.052). We also found that the increase in the number of surgeries performed for AWD closure (when silo placement was needed/complex gastroschisis was present) was associated with the increase in the number of NI. Other studies demonstrated that sepsis was an independent iatrogenic factor for mortality in neonates with AWD, and independent risk factors for surgical site infection included silo placement and prosthetic patch closure [6,23].

In the case of AWD, the last decade has seen a high survival rate in our hospital (as we have demonstrated in previous studies), but also an extension of the average length of hospitalization. This is due to the fact that some of these patients with complex gastroschisis or omphalocele associated with other congenital anomalies died quickly after birth in the first decades analyzed (the median age at death was 5 versus 10 days, *p* = 0.028), thus shortening the hospitalization period [7,18].

A 1990 study conducted in a pediatric surgical service, which overlaps the first period of our study, showed that the largest group of wound infections in children followed operations on the gastrointestinal tract; *Staphylococcus aureus*, *Escherichia coli*, and *Alpha hemolytic Streptococcus* were the most common wound pathogens [14]. Approximately 50% of NIs in the ICU are caused by aerobic Gram-negative bacilli like *Pseudomonas aeruginosa*, *Enterobacter* species, or *Serratia marcescens*; 35% are caused by Gram-positive cocci, coagulase-negative staphylococci or *Staphylococcus aureus* and resistant enterococci; and almost 15% are caused by *Candida* species [21,24]. Overall rates of infections are two to three times higher for patients in a NICU than in other hospital units, and rates of ventilator-associated pneumonia and primary bacteremia—most cases of which originate from intravascular devices—are 10 times higher [21]. In addition, newborns with AWD are subject to an increased risk of NI and late-onset sepsis because of their underlying disease; because they require surgical interventions, and because sometimes prosthetic materials are needed to close the abdominal defect, they have a prolonged period of antibiotic therapy and parenteral nutrition through a central venous catheter, while intra-abdominal pressure and diuresis are monitored with the help of urinary catheters for a long period of time [7,23,25,26]. Risk factor analysis identifies the circumstances that put a patient at increased risk for NI and leads to the development of adequate preventive strategies. In this regard, we demonstrated the fact that being in blood group 0, showing the presence of anemia and undergoing prolonged hospitalization in a NICU increased the rate of NI. Blood groups and hematologic disorders are frequent targets in epidemiological studies because they are genetically determined traits with known polymorphic expression among individuals and populations [27,28]. A recent meta-analysis confirms the increased susceptibility of patients of blood type 0 to viral infections [29], and another study demonstrated that the oral pathogen *Candida albicans* showed a significantly higher interaction with blood group-0 type buccal epithelial cells relative to other blood groups [30]. Our study demonstrated the increased sensitivity of people in blood group 0 to NI and the significant association of blood group 0 with NI. The increase in the rate of anemia by over 20% can be explained by the longer survival period of patients; in the first period, many newborns died in the first 5 days after birth and due to hemoconcentration, anemia was not diagnosed [18].

The preventative rather than therapeutic approach represents the new paradigm for reducing the rate of neonatal infections in NICUs [20]. It is mandatory that we institute in our hospital an active program for the prevention, control and surveillance of NI; the epidemiologist and infectious disease doctor must collaborate with the neonatologist and the pediatric surgeon, and consider the particularities of the patients hospitalized in the NICU. It is also very important for all health care personnel working in an ICU to be familiar with the hospital’s guidelines for the management of invasive devices and to receive training in understanding the epidemiology of and controlling NI [31]. Identifying the microbial cause of NI allows the epidemiologic tracking of pathogens within an NICU, especially those that spread from patient to patient (*S. aureus*, beta-hemolytic streptococci, enterococci or Gram-negative bacilli); therefore, a well-equipped microbiology laboratory had to be instituted in our hospital. After the December 1989 revolution, the 1990–2000 period was one of transition and preparation for our country for European integration, with this period being also called the “lost decade”. In accordance with the efforts made by our country in the field of health reform and for its adherence to European health norms, in 2001 the measure of collecting microbiological samples was implemented in our hospital for all newborns, at the time of admission to the NICU. However, many aspects of the management of anterior abdominal wall defects were similar in the first two decades analyzed (1990–2009), which is why they were grouped. In 2007, Romania became a member of the European Union, and we chose 2010 (which was after the implementation of strict measures in intensive care units) as a benchmark to demonstrate the effectiveness of these measures. In addition to the significant increase in the rate of antenatal diagnosis, a sign of the technological and scientific progress of the medical system in our country after joining the EU, we also noted the significant increase in the survival rates of newborns with AWD [7,18] and the 1.7-fold decrease in the NI rate after 2010. With the application of appropriate preventive strategies, nosocomial infection rates can be reduced by up to 50%; hand washing, a judicious use of antibiotics and interventions, and proper asepsis during procedures remain the most important practices [10,19]. A recent article written by Treglia et al. in 2022 emphasizes the importance of preventive measures in the field of NI with regard to health issues and medico-legal aspects [32]. Thus, in cases of causation-related complications leading to the onset of a NI, they may represent, under certain circumstances, a source of liability for the healthcare facility, with rising costs for public and private health facilities and insurance companies. This may lead to compensation for the injured patients and their families. Therefore, fault-based liability may arise in the case of non-compliant healthcare facility conduct with the scientifically proven precautions meant to prevent the occurrence of any infection [32].

### Limitations of the Study

Although this is the largest cohort of patients with AWD analyzed in a single study in our country, the generalization of the results to the whole population of Romania or other countries will be carried out with caution. Other limitations of the study are that the review was retrospectively performed and, throughout the three decades, analyzed. Neonatology is an evolving field, and there have been significant changes in practice in recent decades; medical knowledge and access to technology have evolved, and this can lead to the appearance of biases in a retrospective study over a long period. This study covers a period of transition between the almost total absence of preventive measures for NI and the period (after 2010) when strict measures were implemented in NICUs; this led to the appearance of a non-homogeneous group of patients. Many patients had a prolonged period of hospitalization and were infected with several bacteria at once or multiple times in a row, at multiple sites (bronchopneumonia, urinary tract infection, surgical wound infection, etc.). Considering these aspects, a precise statistic is difficult to achieve and more biases can result.

*The strength of the study* is the large cohort of approximately one-fifth of the Romanian newborn population. This is the largest number of newborns that has been analyzed in a single study in our country. Being a single-center study, we benefited from the accessibility of individual-level information about each NI and about patient characteristics, sociodemographic factors and comorbidities, in addition to the uniformity of the applied medical and surgical form of management. In addition to being a retrospective study, this study also has the potential to be a quality improvement study. The baseline data found may aid in forming strategies that can be used to lower the risk of NI in those neonates that need hospitalization in an NICU. The results suggest that NIs in neonates are related more to factors of the ICU than of the patients’ overall physiological statuses (type of malformation, sex, method of surgical management, and the presence of an intestinal atresia, which was not associated significantly with NI). There is hope for reducing NI in the next decades by enhancements in aseptic technique, antibiotic stewardship, hand washing, carrying out surveillance swabs periodically, carrying out decolonization treatment for MRSA-positive asymptomatic babies and advances in the technology of invasive devices. This study also draws attention to the necessity of a NI surveillance system in every country as a source of information to identify those strains of bacteria resistant to usual antibiotic therapy and guide prevention actions. It also provides information for future studies and comparisons through the analysis of NI and morbidity linked to neonates with congenital malformations.

## 5. Conclusions

Although in the past 32 years we have been witnesses to many transformational improvements in the management and survival of neonates with abdominal wall defects, some factors still require special attention for an improved outcome. With a better understanding of the factors associated with NI and the application of rigorous measures of asepsis with regard to surgery and high-risk medical devices, the risk of NI may be greatly reduced.

## Figures and Tables

**Figure 1 healthcare-11-01131-f001:**
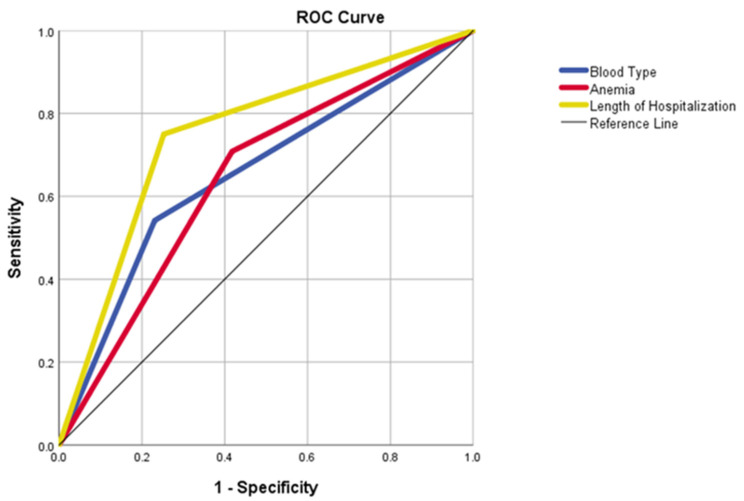
Evaluation of the predictive power of the analyzed variables (blood type 0, LH over 14 days and the presence of anemia) based on the ROC curve (Receiver Operating Characteristic).

**Table 1 healthcare-11-01131-t001:** Patient characteristics in the two analyzed periods.

Patient’s Demographics	Overall	1990–2009	2010–2021	Test Value	*p* Value
**Total patients**	302	199	103		
-Patients with omphalocele -Patients with gastroschisis	139 163	99 100	40 63		
**Median PMA ***** (weeks)	37	38	37	8774 **	0.038
-Patients with omphalocele -Patients with gastroschisis	38 37	38 37	38 37		
**Antenatal diagnostic rate** (%)	21.5	9.0	45.6	7.334 *	<0.01
-Patients with omphalocele -Patients with gastroschisis	18.0 24.5	11.1 7	35.0 52.4		
**Total males** (% of period)	53.0	57.8	43.7	2.327 *	0.020
**Blood type 0** (% of period)	31.8	32.2	31.1	0.193 *	0.847
-Patients with omphalocele -Patients with gastroschisis	47 49	36 28	11 21		
**Median age at death** (days)	5.9	5	10	2225.5 **	0.028
-Patients with omphalocele -Patients with gastroschisis	6.90 5.25	5.8 4.35	10 9.50		
**Median TPN** (days)	4.1	4.1	4.0	9936.5 **	0.664
-Patients with omphalocele	3.0	4.0	3.0		
-Patients with gastroschisis	6.0	5.0	6.0		
**Median length of hospitalization** (days)	13.5	11.0	18.0	7824 **	<0.001
-Patients with omphalocele	12	12	11.5		
-Patients with gastroschisis	17	9	25		
**Nosocomial infection** (number of patients)	102	75	27	1.999 *	0.046
-Patients with omphalocele -Patients with gastroschisis	48 54	38 37	10 17		

* z-test for difference in proportions; ** Mann–Whitney U-test; *** PMA—post menstrual age.

**Table 2 healthcare-11-01131-t002:** Descriptive and inferential statistics of the analyzed patients.

	Overall	1990–2009	2010–2021	Test Value	*p*-Value
Rate of infected patients	33.7%	37.7%	26.2%	1.999	0.046 *
-*Enterobacteriaceae*	20.2%	19.1%	22.3%	0.664	0.507 *
-*Pseudomonas*, *Acinetobacter*	11.3%	14.6%	4.9%	-	0.012 **
-*Staphylococcus aureus*	8.3%	11.6%	1.9%	-	0.003 *
-*Candida*	10.6%	12.1%	7.8%	1.149	0.250 *
-*Enterococcus*	2.0%	1.5%	2.9%	-	0.414 **

* z-test for difference in 2 proportions; ** Fisher’s exact test.

**Table 3 healthcare-11-01131-t003:** Univariate analysis regarding the assessment of the association between the occurrence of NI and the analyzed risk factors.

NI (Yes) vs. Independent Variable:	Omphalocele			Gastroschisis		
	χ^2^	Odds Ratio	*p-*Value	χ^2^	Odds Ratio	*p-*Value
Sex (male)	0.56	1.31	0.452	1.64	1.53	0.201
Number of surgeries (≥2)	7.98	4.00	0.005	8.99	2.82	0.003
Time to surgery (≥6 h)	Not applicable			3.77	1.92	0.052
Intestinal atresia (Yes)	1.41	0.40	0.235	0.63	1.34	0.427
Chromosomal abnormalities (Yes)	4.89	0.36	0.027	*		
Cardiac abnormalities (Yes)	1.48	0.64	0.223	0.20	1.20	0.658
Blood type (0)	13.57	3.94	<0.001	0.08	1.11	0.781
Anemia (Yes)	10.64	3.39	0.001	14.86	4.56	<0.001
Blood transfusions (Yes)	2.31	1.73	0.128	3.22	1.98	0.073
Acute renal failure (Yes)	0.19	1.21	0.665	5.11	2.17	0.024
Conservative treatment	1.77	1.61	0.183	Not applicable		
TPN (>4 days)	3.34	1.94	0.068	5.92	2.37	0.015
Length of hospitalization (≥14 days)	31.81	8.87	<0.001	11.74	3.46	<0.001
Period of time (1990–2009/2010–2021)	2.26	1.87	0.133	1.75	1.59	0.186

* Because in case of Gastroschisis the association of Chromosomal abnormalities is almost inexistent, we didn’t perform the analysis; TPN = total parenteral nutrition

**Table 4 healthcare-11-01131-t004:** Multivariate analysis of the risk of NI occurrence.

Predictor	B	S.E.	Wald	*p-*Value	Odds Ratio	95% C.I. for EXP (B)	
						Lower	Upper
Blood Type (0)	1.356	0.446	9.269	0.002	3.882	2.621	7.297
Anemia (Yes)	0.926	0.451	4.219	0.040	2.525	1.943	4.113
LH (≥14)	1.913	0.439	18.983	0.000	6.772	4.864	8.009

Note. Estimates represent the log odds of “Infected = Yes” vs. “Infected = No”; LH—length of hospitalization.

## Data Availability

The data presented in this study are available on request from the corresponding author.

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
