# Peer review of "Nosocomial Infections Affecting Newborns with Abdominal Wall Defects"

_healthcare, 2023, doi:10.3390/healthcare11081131_

Round 1

Reviewer 1 Report

The paper is well written, the topic is interesting, however there are few improvement to be made.

Line 2 (TITLE) the title is repetitive, it's unnecessary to specify "neonatal health". The title can be modified in "Nososcomial infections affeting newborns with abdominal wall defects"

The "Discussion" can be improved with an insight about medical legal aspects of nosocomial infection. I recommend this interesting scientific paper about the analysis of the juridical and medico legal aspects of healthcare associated infections. 

Treglia M, Pallocci M, Passalacqua P, Sabatelli G, De Luca L, Zanovello C, Messineo A, Quintavalle G, Cisterna AM, Marsella LT. Medico-Legal Aspects of Hospital-Acquired Infections: 5-Years of Judgements of the Civil Court of Rome. Healthcare (Basel). 2022 Jul 18;10(7):1336. doi: 10.3390/healthcare10071336. PMID: 35885861; PMCID: PMC9322800.

Author Response

Dear Reviewer,

Thank you very much for evaluating our manuscript. Your recommendations and comments have helped us greatly improve our manuscript. Here we provide the requested corrections and address the comments.  The changes we have made in the manuscript are highlighted in red.

  1. The paper is well written, the topic is interesting. Line 2 (TITLE) the title is repetitive, it's unnecessary to specify "neonatal health". The title can be modified in "Nososcomial infections affeting newborns with abdominal wall defects"

Response: we corrected the title.

  1. The "Discussion" can be improved with an insight about medical legal aspects of nosocomial infection. I recommend this interesting scientific paper about the analysis of the juridical and medico legal aspects of healthcare associated infections. Treglia M, Pallocci M, Passalacqua P, Sabatelli G, De Luca L, Zanovello C, Messineo A, Quintavalle G, Cisterna AM, Marsella LT. Medico-Legal Aspects of Hospital-Acquired Infections: 5-Years of Judgements of the Civil Court of Rome. Healthcare (Basel). 2022 Jul 18;10(7):1336. doi: 10.3390/healthcare10071336. PMID: 35885861; PMCID: PMC9322800.

 Response: Yes, the recommended paper is very interesting and we added few phrases in the Discussion section regarding the medical legal aspects of nosocomial infection. We also added the mentioned article in our Reference section. Thank you for your advice.

Reviewer 2 Report

This study shows tremendous effort in statistical analysis, data-keeping, and the total number of study subjects. The efforts are commendable. Here are some questions and comments. 

#39, #41: p-value can not equal 0.00. Please include a number or state <0.05 or <0.01

#113: 

a) Line #113 Mentions that surveillance swabs were obtained on admission since 2001, #148 statistical analysis carried out in 2 parts 1990-2009 and 2010-2021

b) The reason for the year 2010 to be chosen as a cut-off period is elucidated in #342. However, that still needs to explain how 1990-2009, the first two decades, were combined, primarily when a new policy was instituted in 2001, in the first decade of study. 

c) What were the surveillance swabs specifically for? 

d) Were they obtained only at admission or at regularly scheduled intervals? For instance, most NICUs dealing with surgical pathology in the United States do weekly MRSA swabs. 

#187: Is the p-value only for the second time period? It is confusing the way it appears in table #2. 

Table 2: What species of Staphylococcus was it? If Staphylococcus aureus, please identify MRSA, MSSA, CONS, etc. Line #302, concerning the 1990 study performed by the pediatric surgical service, mentions Staphylococcus aureus but no more details. 

Table 3 only includes patients from 1990-2009

#351 largest ‘lot’ - largest ‘cohort’ might be a better description or a different choice of words to describe study subjects. 

Further epidemiological details, such as the occurrence of abdominal wall defects together and as gastroschisis and omphalocele separately, in your NICU may provide better insight. 

A breakdown and comparison of nosocomial infections in non-surgical patients, surgical patients without abdominal wall defects, and patients with abdominal wall defects may further solidify information. 

The years 1990-2021 cover over three decades. Neonatology is an evolving field, and there have been significant changes in practice in the last few decades. What other policy changes may impact study results? For instance, antibiotic stewardship, hand washing, surveillance swabs- if done periodically, in addition to what decolonization treatment, if any, is offered for MRSA-positive asymptomatic babies, etc. 

In addition to a retrospective study, this also has the potential to be a quality improvement study. 

Author Response

Dear Reviewer,

Thank you very much for evaluating our manuscript. Your recommendations and comments have helped us greatly improve our manuscript. Here we provide the requested corrections and address the comments.  The changes we have made in the manuscript are highlighted in red.

This study shows tremendous effort in statistical analysis, data-keeping, and the total number of study subjects. The efforts are commendable. Here are some questions and comments.

  1. #39, #41: p-value can not equal 0.00. Please include a number or state <0.05 or <0.01

Response: We corrected.

  1. a) Line #113 Mentions that surveillance swabs were obtained on admission since 2001, #148 statistical analysis carried out in 2 parts 1990-2009 and 2010-2021.  b) The reason for the year 2010 to be chosen as a cut-off period is elucidated in #342. However, that still needs to explain how 1990-2009, the first two decades, were combined, primarily when a new policy was instituted in 2001, in the first decade of study.

Response: After the December 1989 revolution, Romania became integrated into the Western system, becoming a member of NATO in 2004 and the European Union in 2007. The period 1990-2000 was a period of transition and preparation of our country for European integration, this period being also called the "lost decade". In accordance with the efforts made by our country in the field of health reform and for adherence to European health norms, in 2001 the measure of collecting microbiological samples was implemented in our hospital for all newborns, at the time of admission to the NICU.  However, many aspects of the management of anterior abdominal wall defects were similar in the first two decades analyzed (1990-2000 and 2001-2009), which is why they were grouped. As we already explained in the Discussion section, in 2007, Romania became a member of the European Union and we chose 2010 (after the implementation of strict measures in intensive care units) as a benchmark to demonstrate the effectiveness of these measures. Besides the significant increase in the rate of antenatal diagnosis, a sign of the technological and scientific progress of the medical system in our country after joining the EU, we also noted the significant increase in the survival rates of newborns with AWD.

We added a part of this explanation in the Discussion section.

3. c) What were the surveillance swabs specifically for?

d) Were they obtained only at admission or at regularly scheduled intervals? For instance, most NICUs dealing with surgical pathology in the United States do weekly MRSA swabs.

Response: c) Surveillance skin, throat and rectal swabs were obtained on admission of each neonate in our intensive care unit. This is a rule of our hospital because all newborns hospitalized in the NICU come by transfer from other children's hospitals or maternity hospitals. This rule was proposed by the epidemiological service to find out if the newborns came already contaminated or contacted NI in our hospital.

d) For the whole period, every time an infection was suspected (surgical wound with inflammatory phenomena or purulent secretions, purulent tracheobronchial secretions, cloudy urine, surgical re-intervention on the abdomen, fever), secretions were collected, the central venous catheter was changed and given for performing antibiogram, urine was collected for urine culture or blood for blood culture. We did not perform routinely swabs.

  1. #187: Is the p-value only for the second time period? It is confusing the way it appears in table #2.

Table 2: What species of Staphylococcus was it? If Staphylococcus aureus, please identify MRSA, MSSA, CONS, etc. Line #302, concerning the 1990 study performed by the pediatric surgical service, mentions Staphylococcus aureus but no more details.

Response: For better clarity, we modified table 2. The statistical comparison is made between the two periods, and we mentioned this in the text.

Regarding the species of Staphylococcus, they were mostly Staphylococcus aureus. Due to the long analyzed period and the difficult access to patient data in the first decade (patient data were collected from observation sheets in physical - printed format), it was difficult to identify the exact type of Staphylococcus, respectively MRSA, MSSA, CoNS.

We added a statement regarding this aspect in the Limitation section.

  1. Table 3 only includes patients from 1990-2009.

Response: In table 3 we analyzed the potential influence of 13 variables of each surgical condition on the risk of NI appearance.  One of the variables analyzed was the time period. We corrected in the table.

  1. #351 largest ‘lot’ - largest ‘cohort’ might be a better description or a different choice of words to describe study subjects.

Response: We corrected, thank you.

  1. Further epidemiological details, such as the occurrence of abdominal wall defects together and as gastroschisis and omphalocele separately, in your NICU may provide better insight.

Response: Regarding the epidemiological aspects, as well as the treatment of anterior abdominal wall defects, omphalocele and gastroschisis separately, in previous years we published two other studies. (Ţarcă E, Roșu ST, Cojocaru E, Trandafir L, Luca AC, Lupu VV, Moisă ȘM, Munteanu V, Butnariu LI, Ţarcă V. Statistical Analysis of the Main Risk Factors of an Unfavorable Evolution in Gastroschisis. J Pers Med. 2021 Nov 9;11(11):1168. doi: 10.3390/jpm11111168. PMID: 34834520; PMCID: PMC8619615.

Ţarcă E, Cojocaru E, Trandafir LM, Luca AC, Tiutiucă RC, Butnariu LI, Costea CF, Radu I, Moscalu M, Ţarcă V. Current Challenges in the Treatment of the Omphalocele-Experience of a Tertiary Center from Romania. J Clin Med. 2022 Sep 27;11(19):5711. doi: 10.3390/jcm11195711. PMID: 36233585; PMCID: PMC9573750.)

  1. A breakdown and comparison of nosocomial infections in non-surgical patients, surgical patients without abdominal wall defects, and patients with abdominal wall defects may further solidify information.

Response:  Thank you for the suggestion, we will try to make these aspects the subject of future studies.

  1. The years 1990-2021 cover over three decades. Neonatology is an evolving field, and there have been significant changes in practice in the last few decades. What other policy changes may impact study results? For instance, antibiotic stewardship, hand washing, surveillance swabs- if done periodically, in addition to what decolonization treatment, if any, is offered for MRSA-positive asymptomatic babies, etc.

In addition to a retrospective study, this also has the potential to be a quality improvement study.

Response: Thank you for these comments. We have added some of these ideas to the Limitations/Strengths chapter of our manuscript.

Reviewer 3 Report

This paper is extremely important data and should help both within Romania and outside of Romania to determine factors to be addressed to decrease the rate of NI and overuse of antibiotics in neonates with AWD. 

It would be very useful to know the type of nosocomial infections your patients had i.e number with sepsis/bacteremia; number with urinary tract infections; number with pneumonia and what percentage of those where ventilator associated pneumonias; number with central line infections, number with wound infections etc. The authors appropriately state in their discussion “For a newborn with omphalocele or gastroschisis, the occurrence of an infection at the level of the operative wound, due to a prolonged period of mechanical ventilation, TPN or due to invasive maneuvers (central venous catheter) can lead to sepsis, multiple-organ dysfunction syndrome, and directly or indirectly lead to an unfavorable evo- 267 lution, including death [21]. “ but then don’t include the data from their patient on any of these except days on TPN.  This additional data either needs to be included or if not available stated as not available and listed as a significant limitation.

Would also be helpful to comment on why the length of hospital stay was longer more recently.  For many conditions the opposite has been the case i.e.  shorter stays for children with osteomyelitis, prematurity, etc than had previously been the case. 

Minor point but probably don’t need this level of statistical detail in the discussion. “Using a logistic regression model for omphalocele, we found an increased risk of NI for patients with blood group 0 (OR = 3.8, = 0.02), in the case of patients with LH ≥ 14 days (OR = 6.7, < 0.01) and in the presence of anemia (OR = 2.5, = 0.04).” 

Author Response

This paper is extremely important data and should help both within Romania and outside of Romania to determine factors to be addressed to decrease the rate of NI and overuse of antibiotics in neonates with AWD.

Response: Dear Reviewer, thank you for your time evaluating our manuscript. Your recommendations and comments help us improve our manuscript. Here we provide the requested corrections and address the comments. The changes we have made in the manuscript are highlighted in red.

  1. It would be very useful to know the type of nosocomial infections your patients had i.e number with sepsis/bacteremia; number with urinary tract infections; number with pneumonia and what percentage of those where ventilator associated pneumonias; number with central line infections, number with wound infections etc. The authors appropriately state in their discussion “For a newborn with omphalocele or gastroschisis, the occurrence of an infection at the level of the operative wound, due to a prolonged period of mechanical ventilation, TPN or due to invasive maneuvers (central venous catheter) can lead to sepsis, multiple-organ dysfunction syndrome, and directly or indirectly lead to an unfavorable evolution, including death [21]. “ but then don’t include the data from their patient on any of these except days on TPN. This additional data either needs to be included or if not available stated as not available and listed as a significant limitation.

Response:  Dear reviewer, you are right to request this important information. Due to the long analyzed period and the difficult access to patient data in the first decade (patient data were collected from observation sheets in physical - printed format), sometimes it was difficult to identify the exact source of the infection. Also, many patients had a prolonged period of hospitalization and were infected with several bacteria at once or in a row, at multiple sites (bronchopneumonia, urinary tract infection, surgical wound infection, etc.). Considering these aspects, a precise statistic is difficult to achieve, and we opted to group all these infections in the "nosocomial infections" category. A total of 33.7 % patients were infected with one or more of the specified species of bacteria or fungi.

We added a statement regarding these aspects in the Limitation section.

  1. Would also be helpful to comment on why the length of hospital stay was longer more recently. For many conditions the opposite has been the case i.e.  shorter stays for children with osteomyelitis, prematurity, etc than had previously been the case.

Response: In the case of AWD, the last decade has brought a higher survival rate in our hospital (as we have demonstrated in previous studies), but with the extension of the average length of hospitalization. This is due to the fact that some of these patients with complex gastroschisis or omphalocele associated with other congenital anomalies died quickly after birth in the first decades analyzed (median age at death was 5 versus 10 days, p = 0.028), thus shortening the hospitalization period.

We added this statement in the Discussion section. Thank you.

  1. Minor point but probably don’t need this level of statistical detail in the discussion. “Using a logistic regression model for omphalocele, we found an increased risk of NI for patients with blood group 0 (OR = 3.8, p = 0.02), in the case of patients with LH ≥ 14 days (OR = 6.7, p < 0.01) and in the presence of anemia (OR = 2.5, p = 0.04).”

Response: we corrected.

Round 2

Reviewer 3 Report

Authors have addressed my questions appropriately.  Should this be done again prospectively would suggest collecting the following data type of nosocomial infections your patients had i.e number with sepsis/bacteremia; number with urinary tract infections; number with pneumonia and what percentage of those where ventilator associated pneumonias; number with central line infections, number with wound infections etc